# Influence of Sports Activities on Prosocial Behavior of Children and Adolescents: A Systematic Literature Review

**DOI:** 10.3390/ijerph19116484

**Published:** 2022-05-26

**Authors:** Jiayu Li, Weide Shao

**Affiliations:** College of Physical Education and Health Sciences, Zhejiang Normal University, Jinhua 321004, China; jiayu1001@zjnu.edu.cn

**Keywords:** prosocial behavior, sports activities, children and adolescents, review

## Abstract

Prosocial behavior plays a key role in interpersonal relationships during the growth of children and adolescents. Good prosocial behavior is the foundation for the healthy development of children and adolescents. In recent years, the role played by some sports activities in children and adolescents’ prosocial behaviors has attracted much attention. However, the effects of physical activity on prosocial behavior have not been summarized. Objective: We aimed to clarify the role of sports activities in children and adolescents’ prosocial behaviors. Methods: We searched databases for 27 interventional studies on the influence of sports activities on children and adolescents’ prosocial behaviors published in peer-reviewed English journals. Subsequently, inductive, summary, analytical, and evaluation methods were used to systematically analyze and evaluate the literature. Results: Sports activities can improve children and adolescents’ prosocial behaviors. Different sports activities also influence children and adolescents’ prosocial behaviors differently. Moreover, sports activities can improve the prosocial behaviors of children and adolescents with special educational needs. Conclusion: This review demonstrates that sports activities improve the prosocial behavior of children and adolescents. At the same time, we find that children and adolescents with special educational needs should be allowed to participate in more sports activities.

## 1. Introduction

Prosocial behavior refers to a kind of behavior that conforms to social hope and has no obvious benefit to the actor, but the actor voluntarily brings benefits to the recipient of the behavior. In fact, it is a series of positive social behaviors, in which individuals voluntarily provide help or benefit others and society. It is characterized by acts of helping, sharing and comforting [1]. Currently, the causes of prosocial behavior are still controversial and two major theories are frequently discussed: instinct and acquisition theories [2]. Instinct theory illustrates that prosocial behavior is ingrained in human genetic material and is one of the species’ instinctive mechanisms [3]. Acquisition theory demonstrates that prosocial behavior evolves from continuous learning during individual socialization [4]. As an important part of individual socialization, prosocial behavior has appeared at an early age of human life and is promoted with the process of individual socialization [5,6,7,8,9]. Essentially, prosocial behavior is an important fundamental positive social behavior in which people can form and maintain good relationships as well as enhance social harmony [10,11,12,13]. Notably, compared to healthy children, children with special educational needs experience inferiority complexes and introversion, leading to insufficient prosocial behavior [14,15]. The factors influencing prosocial behavior include internal and external aspects. Internal factors include biology, individual cognition, self-concept, and personality, whereas external factors include family, society and school [16,17]. In the context of the declining global fertility rate, studies found fewer prosocial behaviors in children and adolescents [18]. Consequently, children and adolescents with social problems have increased, such as children’s antisocial behaviors, teenagers’ addiction behaviors and crimes, which cause great harm to themselves, others and social stability [10,19,20]. Methods of improving the prosocial behavior of children and adolescents have become an urgent issue for psychologists and educators [21]. Psychotherapy, reading therapy, family therapy and school intervention are the main strategies to improve prosocial behaviors for children and adolescents [22,23,24]. However, these strategies have achieved little effect due to material costs, time, space and implementation difficulty. Therefore, other effective and convenient strategies still need to be investigated to improve prosocial behavior.

Physical activity represents any form of body movement with physical exertion promoting health including active recreational play and sports, which can be played at any level of skill [25]. In addition, physical activity can be performed individually (judo, tennis, or boxing, etc.) or as a team (football, rugby, basketball, volleyball, handball, etc.) [26,27]. Physical activity plays an important role in developing fitness and mental health in children and adolescents. Physical activity results in a series of positive adaptions in physiological responses by mainly improving the cardiovascular and muscular systems. Furthermore, physical activity enhances psychological variables of mental health and helps children and adolescents actively integrate into society [17,28]. Rodríguez (2016) et al. found that children showed that physical activity reduced aggressive behavior in children and adolescents, suggesting increasing their cooperation and sharing qualities [29]. As supportive, Lee et al. (2017) found that physical activity can improve an individual’s social skills and problem-solving abilities [30]. In addition, Moeijes (2018) et al. showed that children and adolescents who actively participate in physical activities develop better prosocial behaviors and fewer interpersonal problems [31]. A series of studies illustrate that an association probably exists between physical activity and prosocial behavior in children and adolescents. Therefore, physical activity may be widely promoted as a potential strategy. The effects of physical activity on prosocial behavior have become a popular scientific topic in sports psychology. However, no research summarizes and discusses physical activities as an intervention for prosocial behavior. Therefore, the purpose of this literature review was to summarize the intervention research on physical activity on children and adolescents’ prosocial behavior.

## 2. Method

We adopted the methods of systematic review and meta-analysis for this study.

### 2.1. Search Strategy

We adopted a two-step search strategy to determine relevant research. Firstly, we determined the database search. We selected four English electronic science databases for complete and thorough retrieval, namely Pub-med, Eric, Psychology and Behavioral Sciences Collection, and Web of Science. Secondly, the search strategy included a combination of the following keywords: (1) sports movement OR physical exercise OR sports activities OR sport* OR motor OR athletic sports; (2) child OR children OR childhood OR pediatric OR infant OR kids OR adolescent* OR teens OR teenager* OR juvenile OR school-aged children; (3) altruism OR humanitarianism OR prosocial behavior OR prosocial behavior OR social behavior OR compassion OR help* OR care OR caring OR empath* OR shar* OR donat* OR comfort* (see the Appendix A (Appendix A) for the search strategy used for each database). All the reference lists included in our research were searched manually to determine other related papers.

### 2.2. Study Selection

A total of 8374 related studies were obtained through an initial search of the database. Firstly, the articles are imported into the literature management software Endnote, and 3642 articles were obtained after the duplicate articles were removed. After reading the title and abstract and excluding the non-full-text, 327 articles were initially obtained. Subsequently, the rest of the articles were reviewed by full-text reading, and 25 full-text articles were obtained after excluding irrelevant articles. In addition, two related articles were supplemented by manual retrieval of reference articles. Finally, 27 articles were included for systematic review and analysis (Figure 1).

### 2.3. Inclusion and Exclusion Criteria

Our study included research on the influence of physical activity on children and adolescents’ prosocial behaviors, including sports activities, physical exercise, and physical education, while excluding research unrelated to the topic. The literature in our study included intervention research and excluded observational research. Children and adolescents aged 3–18 were included, excluding other age groups. Published articles in English excluded books, chapters in books, unpublished papers, conference papers, doctoral dissertations, or non-English articles.

### 2.4. Data Extraction

Data were extracted from each study using a general form that included the first author’s name, publication year, research purpose, research results, participant characteristics, grouping type, research design, intervention measures, and detection tools (Table 1).

### 2.5. Quality Assessment

We evaluated the included literature using the PEDro scale, a credible rating scale developed by the Centre for Evidence-based Practice in Australia [28]. The scale includes random grouping (two items), blind method (three items), data reporting items (three items), data analysis (one item) and follow-up (one item), with a total of ten criteria. Each item was recorded as 1 point when it appeared in an article, and 0 points when it was not reflected, with a total score of 0–10 points. In order to avoid subjective opinions, two reviewers evaluated the opinions and a third reviewer judged the differences (Table 2).

### 2.6. Data Syntheses and Analysis

No meta-analysis was conducted since many intervention methods were used in the study. Instead, the researchers used the best evidence synthesis method [58]. This method has been widely accepted and used in review articles in the same field [59]. We considered the consistent results of these high-quality studies as strong evidence. We considered the generally consistent results of two high-quality studies as moderate evidence. We considered the conclusion of a high-quality study as limited evidence. We considered the contradictory results of the research as mixed evidence. We considered consistent results in many low-quality studies as insufficient evidence. Consistency means that more than two-thirds of the research results follow the same direction.

## 3. Results

### 3.1. Study Description

The amount of published correlation literature. A systematic search was conducted on the relevant literature published as of April 2022 and 27 articles were retrieved, including one article in 2013, two in 2015, nine in 2016, five in 2017, one in 2018, two in 2019, four in 2020 and three in 2021. The publication of articles first declined and then increased (Figure 2a).

Authors’ region and country of origin. As shown in Figure 2b,c, the 27 relevant research papers included in this study are from the United States, China, Korea, Turkey, Lithuania, Spain, Italy, the Netherlands, Switzerland, Serbia, Germany, Brazil, Egypt, Czechoslovakia, Iran, and India. Geographically, Asian countries published 10 articles, accounting for 37% of the total, European countries published 10 articles, accounting for 37% of the total, North America published 5 articles, accounting for 19%, Africa published 1 article, accounting for 3% and South America published 1 article, accounting for 3%. (Figure 2b,c).

Design and research methods. As shown in Table 1, there are 27 articles in total, and the research design is interventional. In the sampling survey, the types of 22 articles are randomly sampled, accounting for 81% of the total literature; the rest of the articles are not randomly sampled (Table 1).

The source of the subject. Among the 27 articles included in this study, the subjects ranged in age from 3 to 18 years old. Among the 27 articles, 16 articles included children and adolescents with good physical and mental health, 11 articles included children and adolescents with special educational needs.

Intervention factors and measurement tools. The intervention factors mentioned in the 27 articles included in this study involved sports activities. Each center recorded measurements by quoting or developing questionnaires and scales according to specific content to measure related indicators.

Quality of articles. The quality of the 27 methods included in the study is shown in Table 2. The PEDro scores of these studies ranged from 2 to 9. According to the evaluation scale, 17 studies scored five points or higher; therefore, they were considered high quality. A total of 10 studies scored less than five points. The main areas of weakness in Chinese jurisprudence included in this study are related to blind technology. Most of the studies are single-blind experiments, 20 of them scored 1 for blinding technology, only 3 scored 3, and 4 studies scored 0.

This study first described the influence of sports activities on children’s prosocial behavior. Then, it explained the influence of different sports activities on prosocial behavior. Finally, it expressed different sports activities on prosocial behavior between healthy children and children with special educational needs.

### 3.2. Sports Activities Can Improve the Prosocial Behavior of Healthy Children and Adolescents

Sixteen studies have reported the influence of sports activities on healthy children and adolescents’ prosocial behaviors. Six of these were through the intervention of sports activities in physical education class (including running, jumping, ball games, etc.) [39,45,47]. They found that sports activities positively influence children and adolescents’ prosocial values [29,60,61,62]. Students who participate in sports or social activities have a higher sense of psychological well-being and mission than students who do not participate in any sports activities [30,44]. Aksoy’s research proved that sports activities in the classroom are a useful tool for the moral development of children and adolescents [63]. Lee’s research found that sports activities help participants become more active, have better social skills, and become better problem-solvers [30]. In addition, Park (2017) found that children from multicultural backgrounds facing foreign cultures and unfamiliar environments may experience cultural adaptation pressures and problems such as alienation, anxiety, depression, and sleep disorders. Participating in PEC activities for eight weeks reduced children’s aggression and stress levels in multicultural families. Their social skills and physical health also improved [47].

The other five were by implementing precise teaching plans in physical education classes. Our research found that children and adolescents’ prosocial behaviors significantly improved by implementing education (Olympic and EPHECT programs) and training programs (two-hour weekly training) [35,46,55]. Students participating in the training program demonstrated better communication, cooperation, and social adaptation [43,64]. The education and training program further demonstrated how educators in physical education classes could turn psychological theory into practice. According to the best evidence synthesis, these research results strongly affirm the positive influence of sports activities on children and adolescents’ prosocial behaviors.

### 3.3. Different Sports Activities Have Different Effects on Children’s and Adolescents’ Prosocial Behaviors

Among the 27 included studies, 14 specific sports activities were identified for improving the development of children and adolescents’ prosocial behavior. These sports include parkour running [36], football [40,65], table tennis [38], martial arts [42], squash ball [42], table tennis [41], yoga [37], dance [66], basketball, volleyball, frisbee [32], skiing and cycling [33], hydro gymnastics [34] and brain gym exercise [57]. Specifically, students who participate in parkour emphasized that parkour activities help them improve their social skills and problem-solving abilities [36]. Female participants also described the benefits of urban dance, including positive identity and values [66]. Students who participate in football activities reported improved self-control, respect, and cooperative behavior [40,65]. Participants in yoga [37], martial arts [42], table tennis [41] and squash [42] demonstrated an increase in positive social behaviors. In other studies, high-intensity interval training [45] and stress training [67] have also benefited the development of children’s prosocial behavior. Out of these eight studies, six were cooperative activities, and two were solo sports. The results show that children who participate in team sports, outdoor sports or competitions show fewer problems than those who participate in individual sports, indoor sports, or simple training [31,48]. Individual athletes are more likely to report anxiety and depression than team athletes [68]. Group exercise and team exercise have also been improved results related to sports activities more than individual exercise [69].

### 3.4. Sports Activities Can Improve Prosocial Behaviors of Children and Adolescents with Special Educational Needs

Out of the 27 studies, 11 reported the effects of sports activities on the prosocial behaviors of children and adolescents with special educational needs. Eight articles discussed neurodevelopmental disorders (four articles on ADHD, attention deficit and hyperactivity disorder; four articles on ASD, autism), two articles on intellectual disability (Down syndrome), and one article on physical disability. Pan (2016) indicated that long-term physical exercise improves the cognitive function of ADHD children and reduces attention and thought problems in ADHD children [38]. Messler (2018) subsequently confirmed this conclusion for high-intensity interval training [48]. It has been suggested that sports intervention should be used as a treatment for preschool children with ASD [42,51,52]. Researchers also found that adaptive football training can reduce the aggression of Down syndrome (DS) teenagers, improve their social behavior, and teach them simple motor skills [50,56]. Studies have also shown that physical exercise reduces the socially distant, withdrawn, and isolated behavior of children and adolescents with special educational needs [54]. However, Maremka’s research shows that school-based sports programs do not affect adolescents with physical disabilities’ mental and social health or attention [53].

## 4. Discussion

The purpose of this systematic review was to summarize the current impact of sports activities on children and adolescents with prosocial behaviors. The final analysis includes 27 interventional studies from 16 countries. Seventeen studies are high-quality, and there are good reasons to believe that the influence of sports activities on children and adolescents’ prosocial behavior is important. On the basis of strictly limiting the nature of the included research to be interventionist, as described in Figure 1, the years of the literature included in this research span from 2013 to 2022. It can be guessed that from 2013, both psychologists and educators gradually discovered the influence of sports participation on children and adolescents’ prosocial behavior. In addition, from the region and country where the literature was published, the related research focused on developed countries and some developing countries. This may be because, with the increase in material wealth, the socialization of children and adolescents has attracted more attention. Judging from the amount of published related literature, the attention paid to the influence of sports activities on children’s pro-social behaviors and teenagers’ pro-social behaviors is still scarce in the world. Therefore, we hope that this research can arouse more regional and national researchers’ attention to the influence of sports activities on children’s pro-adolescent and pro-social behaviors. At the same time, more randomized controlled trials (RCT) with high quality and strong evidence have been carried out to further demonstrate and clarify the positive effects of physical activities on children and adolescents’ prosocial behaviors.

### 4.1. Sports Activities Can Improve the Prosocial Behavior of Healthy Children and Adolescents

In this systematic review, 16 high-quality studies confirmed the influence of sports activities on the prosocial behaviors of healthy children and adolescents. These study results that proper physical activity intervention for children and adolescents positively regulates the influence of prosocial behavior [46]. This result is consistent with the previous analysis, which reported the significant impact of sports activities on children and adolescents as individuals in society [28]. The influence of sports activities on children and adolescents’ prosocial behavior has two explanations. Firstly, sports activities contribute to children and adolescents’ healthy and active lifestyles. Participating in sports has many benefits for physical and mental health. Secondly, sports activities serve as a communication tool for developing other life skills, such as social and coping skills [30]. Sports activities provide a suitable and flexible environment, helping children and teenagers interact with all kinds of people and actively explore various life skills, such as teamwork [70], problem-solving [39], and goal setting [71]. The above results strongly prove that sports activities have a positive intervention effect on improving children and adolescents’ prosocial behaviors. When children and teenagers participate in sports activities, they treat their teammates in a more prosocial way and are also recipients of other people’s prosocial behaviors. This finding is consistent with the concept of social identity theory, where individuals are driven to expand groups by treating group members more favorably [72]. Considering the influence of physical activity classes on children and adolescents’ prosocial behavior, teachers can better understand the characteristics of physical activity intervention through theory or teaching methods and help improve children and adolescents’ non-motor (cognitive, social, and emotional) skills. Here, some key factors are emphasized. Teachers’ participation in organized sports activities and intervention in group activities creates an incentivized atmosphere that supports children and adolescents’ active exploration [73].

### 4.2. Different Sports Activities Have Different Effects on Children’s and Adolescents’ Prosocial Behaviors

Eight high-quality studies confirmed that different sports interventions have different effects on children and adolescents’ prosocial behaviors. Firstly, they affirm the extraordinary role of team sports. Participating in team sports is more likely to be positively affirmed by peers than participating in individual sports. Positive affirmations from peers cultivate self-esteem and develop mutual friendships, thus promoting the development of prosocial behavior [31]. Secondly, cooperative sports activities improve motivation, enhance self-efficacy, promote continuous play, and enhance prosocial behavior. It has been widely recognized that competitive games increase children’s aggression [69]. However, a recent study on competitive activities found that game competition helps motivate children to adjust to unpleasant emotions such as depression, anger, and jealousy, reducing their behavioral problems and improving relationships with their peers [74]. Competitive activity brings excitement and freshness to children and teenagers, enabling them to concentrate. These activities are pleasant and further stimulate prosocial behavior. Ball games with cooperation and competition as two elements are recommended. This article also found that high-intensity interval training improves the creativity, self-control, and social skills of children and adolescents [45]. The reason for this can be found in physiology. High-intensity interval training increases cerebral blood flow and catecholamine production, improving children and adolescents’ self-efficacy, reducing negative emotions, promoting sustained entertainment, and increasing the occurrence of prosocial behaviors [75]. From the cognition and teaching perspective, sports activities improve the discipline and self-awareness of children and adolescents, thus affecting their social norms and behaviors [49]. When high-intensity activities are associated with cooperative activities, high-intensity sports activities in pairs or groups may be a new strategy for improving prosocial behavior [76]. Finally, highly structured stress training has also been found to develop a series of skills related to psychological resilience [67]. To overcome the psychological barriers deliberately set up in adversity training camps, participants challenge their physiological needs by increasing their confidence and ability to defeat opponents. This attitude toward an active response to challenges in the face of difficulty focuses on children and adolescents’ cultivation of psychological willfulness and social adaptability. Stress training can also improve thought processes, enhance self-understanding, and motivation, finally leading to changes in the personality, attitude, and skills. In the future, the relationship between sports activities and children and adolescents’ prosocial behaviors can be further elaborated by integrating different types of innovative game activities. Therefore, sports activities are expected to become an attractive scheme for improving children and adolescents’ prosocial behaviors.

### 4.3. Sports Activities Can Improve Prosocial Behaviors of Children and Adolescents with Special Educational Needs

The influence of sports on children and adolescents is not limited to healthy children; it is more important for children with physical and psychological defects. Social and communication deficiencies in children and adolescents with special educational needs cause them to participate less in sports activities and cooperate with others. Their overall health is affected, and they become further deprived of social adaptability. Through organized sports activities, children and adolescents with special educational needs have the opportunity to communicate with others, which is beneficial to their physical and mental health [42,51]. Sports activities are regarded as an important environment for children and adolescents with special educational needs. They are exposed to a supportive, inspiring, and educational atmosphere. In physical education classes, children and adolescents with special needs can also be organized into sports activities nurturing specific skills, including exercises to train specific life skills [62]. Participation in sports affects the prosocial behavior of children and adolescents with disabilities, helping them reach an appropriate level of excitement through sensory stimulation and adjustment [77]. However, school-based sports programs appear not to affect adolescents with physical disabilities’ mental and social health or attention [53]. The reason for this may be that the frequency and duration of sports intervention are insufficient to influence the prosocial behavior of children and adolescents with special educational needs. Out of the nine studies on disabled children, eight are considered high-quality, and the results are consistent. We affirm that sports activities improve the prosocial behavior of children and adolescents with special educational needs. In the future, sports activities can be used as a feasible treatment for disabled children’s social interaction problems, helping to improve their prosocial behaviors. Meanwhile, sports activities should also be considered a form of intervention in clinical practice.

### 4.4. Research Limitations and Future Prospects

Although this review discusses the influence of sports activities on children and adolescents’ prosocial behaviors from three aspects, its limitations should be examined correctly. This review provides some direction for further research on related content. Although a thorough literature search was conducted for articles published before 2022, some published studies may have been ignored due to different keywords from those used in the current work. Secondly, the sample size of some of the studies was small (85%, 23 studies), limiting the universality of the findings and strong evidence.

Despite the above limitations, this review systematically collates the literature reports on the influence of sports activities on the prosocial behaviors of children and adolescents. Future research can explore high-quality randomized controlled trials and provide conclusive evidence for the influence of sports activities on children and adolescents’ prosocial behaviors. Future research should also focus on developing sports activities that help children and teenagers master life skills and the cognitive process between students’ learning and application of life skills in sports activities. More studies should interview students directly, asking them whether they think these social skills are necessary to their lives to draw a clearer conclusion. Cooperation and competition sports activities, high-intensity training, and adversity training should achieve more results by integrating different innovations. Sports activities are expected to become an attractive alternative for improving children and adolescents’ prosocial behaviors and promoting extensive and active participation. A more comprehensive sports evaluation is needed in the future. Sports activities can be combined with conventional therapy for children and adolescents with special educational needs.

## 5. Conclusions

This review systematically examined existing evidence on the influence of sports activities on children and adolescents’ prosocial behaviors and determined the most useful and powerful. Strong evidence shows that sports activities improve the prosocial behavior of healthy children and adolescents. Different sports activities influence the prosocial behavior of children and adolescents differently. Sports activities can also improve the prosocial behavior of children and adolescents with special educational needs. More randomized controlled trials should be conducted in future research if conditions permit. More convincing evidence should also be obtained based on the scheme’s feasibility and including suitable research populations. This article is aimed at different groups of people and provides appropriate and reasonable sports intervention measures for children and adolescents with prosocial behavior problems. It also contributes to the healthy physical and psychological development of children and adolescents worldwide.

## Figures and Tables

**Figure 1 ijerph-19-06484-f001:**
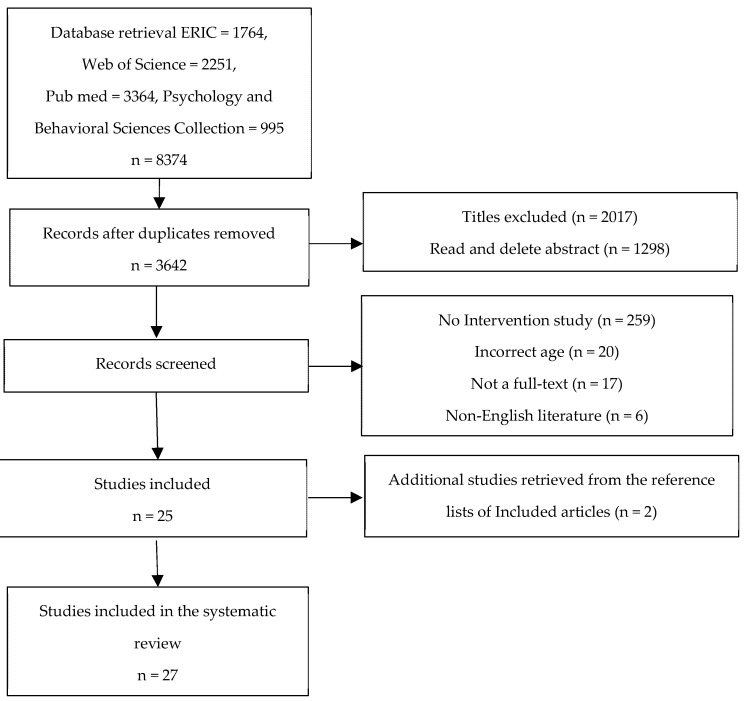
Flow diagram of the study selection process.

**Figure 2 ijerph-19-06484-f002:**
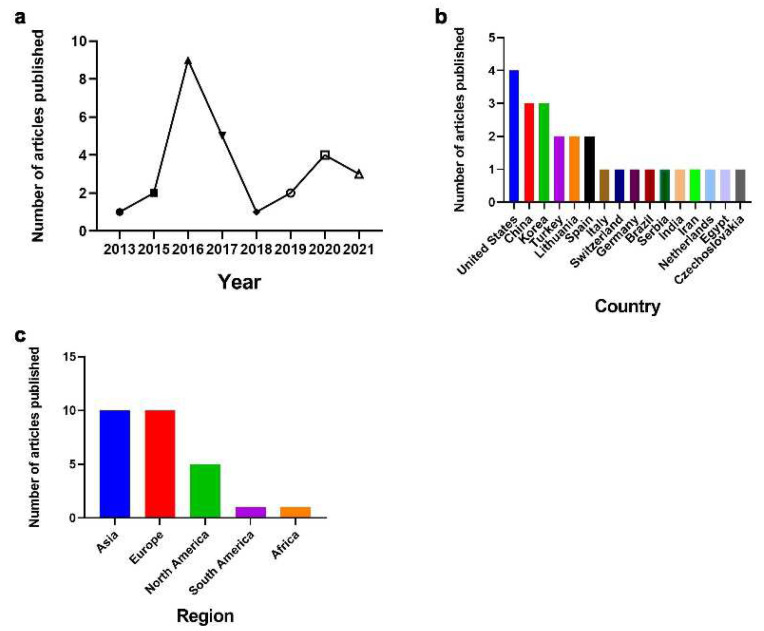
The time and area of publications. (**a**) Year of publications. (**b**) Author’s source region. (**c**) Author’s source country.

**Table 1 ijerph-19-06484-t001:** Study Characteristics.

Author (Year)	Purpose	Sample Characteristics	Sampling Type	Research Design	Intervention Measure	Testing Tool	Result
Samalot-Rivera et al. (2013) [32]	To test the influence of social skill teaching on the social behavior of students with emotional or behavioral disorders in physical education class.	6 students between 10 and 17 years old	random	Intervention	Basketball, Volleyball and Frisbee	unclear	Social skills teaching in physical education class has had a positive impact on students.
Pintérová et al. (2015) [33]	To verify the potential influence of experiential education in physical education classes on social relations of students with behavioral disorders.	54students between 12 and 15 years old	Non-random	Intervention	skiing and cycling	Social Measurement Rating Questionnaire (SORAD)	Students participating in sports activities have improved their influence and popularity.
Metwaly et al. (2015) [34]	To examine the influence of water gymnastics on the sports ability and social behavior of primary school students.	24 children (12 boys and 12 girls)	random	Intervention	Hydro gymnastics	The Social Skills scale	Ten weeks of water gymnastics training can improve pupils′ physical skills and social skills
Sukys et al. (2016) [35]	To study the influence of a comprehensive Olympic education plan on the development of adolescent prosocial behavior.	411 students in Lithuania (182 boys and 229 girls, average age 13.9 years)	random	Intervention	Olympic education project	Prosocial Tendency Scale (PTM-R)	Adolescents’ prosocial behavior has improved significantly.
Río et al. (2016) [36]	Evaluate students’ thoughts, opinions and feelings after a parkour learning unit.	26 sixth grade students aged between 11 and 12 years old	random	Intervention	parkour	Interview	Parkour helps students improve their social skills and problem-solving skills.
Folleto et al. (2016) [37]	To explore the influence of a yoga course in physical education classes on sports ability and social behavior parameters of children.	16 first-year students	random	Intervention	yoga	Children’s perception ability and socially accepted image scale.	Physical yoga practice is helpful to children′s development.
Pan et al. (2016) [38]	The effects of table tennis for 12 weeks on motor skills, social behavior and executive function of children with attention deficit hyperactivity disorder (ADHD) were evaluated.	32 boys aged 6–12 with ADHD	random	Intervention	Table tennis	The CBCL	Long-term physical exercise can significantly improve the cognitive and behavioral functions of ADHD
Gorucu et al. (2016) [39]	To explore the influence of arranging physical education classes by cooperative learning on middle school students′ problem-solving ability.	48 students studying in Konya	random	Intervention	Design of physical education class by Cooperative Learning Mode	Problem Solving Inventory for Children	children and alleviate their social problems.
Ferguson et al. (2016) [40]	To explore whether an SST project in a sports camp (football) environment can improve social skills and sports ability in a short time.	8 boys of 12 years old	random	Intervention	Behavior training intervention	A multiple-baseline design across skills	SST project can improve participants’ social skills and sports ability.
Weiss et al. (2016) [41]	Explore golf as a tool to teach life skills and improve development results.	405 teenagers (301 boys and 104 girls)	Non-random	Intervention	The First Tee course	Self-Perception Profile for Adolescents (SPPA);the Behavioral Conduct subscale; Multidimensional Scales of Perceived Self-Efficacy	The first Tee is effective in teaching life skills and promoting development achievements.
Phung et al. (2016) [42]	To evaluate the effectiveness of MMA intervention in improving school-age boys’ ASD social skills and reducing problem social behaviors.	34 children with ASD (aged 8–11 years, 28 boys, 6 girls)	random	Intervention	Comprehensive fighting (MMA)	The Lifetime Social Communication Questionnaire	The benefits of martial arts training for autistic boys are helpful to solve the physical needs of autistic children.
Lang et al. (2016) [43]	To implement a coping training plan (EPHECT) in general physical education courses and evaluate its influence on coping and stress of higher vocational students.	131 students in eight classes	random	Intervention	(EPHECT) training plan	Coping Questionnaire for Children and Adolescents; the Adolescent Stress Questionnaire (ASQ)	The positive contribution of training project EPHECT to the development of students’ adaptive coping skills
Bakır et al. (2017) [44]	To explore whether sports are effective for mental health and enthusiasm.	60 students in Grade 10	random	Intervention	Organized sports activities and social activities	Psychological Well-being Scale (WEMWBS) and Enthusiasm Scale	Students who participate in sports or social activities have higher scores of psychological well-being than before.
Ruiz-Ariza et al. (2017) [45]	To analyze the influence of cooperative high intensity interval training (C-HIIT) on creativity and emotional intelligence (happiness, self-control, emotion and social ability) of teenagers aged 12–16.	84 teenagers aged 12–16	random	Intervention	High intensity interval training (C-HIIT)	The CREA test	C-HIIT training improves the social skills of teenagers.
Lee et al. (2017) [30]	To explore the influence of extracurricular sports on the development of teenagers’ life skills, and find out which features of the sports will affect their life skills acquisition.	6 children (4 boys and 2 girls)	random	Intervention	A 12-week extracurricular project	Interview	Sports activities make participants more active, have better social skills and even become better problem solvers.
Malinauskas et al. (2017) [46]	Analyze the characteristics of social responsibility education for football sports school students.	52 boys, 26 in the experimental group and 26 in the control group.	Random	Intervention	Educational experiment	Modified Social Responsibility Questionnaire	The sense of social responsibility of sports school students has greatly developed after the education plan.
Park et al. (2017) [47]	To explore the influence of physical education curriculum on children’s aggressiveness, sociality, stress and physical ability.	50 children (25 boys and 25 girls	random	Intervention	Eight-week physical education class	Aggression scale Sociality scale Stress scale	PEC reduced the level of aggression and stress of children from multicultural families, and improved their mental health and social behavior.
Messler et al. (2018) [48]	To compare the influence of high intensity interval training (HIIT) on the social behavior of ADHD boys.	28 boys (8–13 years old)	random	Intervention	High-intensity interval training	the hyperkinetic disorder questionnaire (SBB-HKS) and the KINDL-R questionnaires mental health	HIIT with high intensity interval training improved the physical quality and life quality of ADHD boys.
Gulati et al. (2019) [49]	The purpose of this study is to observe the influence of 4 months of yoga practice on children’s attention, self-esteem and peer interaction.	16 children (78 boys), average age = 10.2 years,	random	Intervention	Practice yoga for 60 minutes every day, seven days a week.	Indian Adaptive Child Self-esteem Scale (SEIC)	Yoga is beneficial to school children and improves their social behavior.
Ryuh et al. (2019) [50]	The effectiveness of a football project in improving the bad behavior of children with intellectual disabilities and non-intellectual disability	20 normal students (average age = 10.9 years old) and 20 students with ID (average age = 10.6 years old,	random	Intervention	Inclusive Football (INS) Program	the Withdrawn Behavior Checklist (WBC) and Social Distance Scale (SDS)	Social distance and withdrawn behavior of intellectually disabled children have decreased.
Cai et al. (2020) [51]	The effects of a long-term exercise intervention on SC and white matter integrity (WMI) of ASD children were evaluated.	29 children diagnosed as ASD by DSM-5	Non-random	Intervention	Mini basketball training program	Social Response Scale (SRS-2), Children Autism Rating Scale (CARS)	Exercise can improve the SC and white matter integrity of autistic children.
Cai et al. (2020) [52]	The influence of the mini basketball training program (MBTP) on the social ability of preschool children with autism spectrum disorder (ASD) was investigated.	59 ASD preschool children aged 3–6 years	Non-random	Intervention	Mini basketball training program	Social Response Scale (SRS-2), Children Autism Rating Scale (CARS)	MBTP can improve the social communication ability of preschool autistic children.
Zwinkels et al. (2020) [53]	To explore whether school-based sports will affect the psychosocial health and attention of physically disabled teenagers.	70 physically disabled children and adolescents (average age 8–19 years old, 54% of whom are boys)	Non-random	Intervention	School-based sports activities	Children’s Self-perception Scale (SPPC)	School sports have no effect on the mental and social health and attention of physically disabled teenagers.
Ringenbach et al. (2020) [54]	To examine the effects of assisted bicycle therapy (ACT) on the adaptability/maladjustment behavior, depression and self-efficacy of adolescents with Down syndrome (DS).	21 teenagers with DS.	random	Intervention	Auxiliary circulation therapy (ACT)	Vineland Adaptive Behavior Scale (VABS) II	DS teenagers’ social coping skills have been improved
Condello et al. (2021) [55]	This paper puts forward a sports intervention method with abundant sports to promote the development of sports, cognition and social emotional skills.	242 fifth-grade students aged 10–11 years	Random stratified sampling	Intervention	Multi-sports intensive exercise	Evaluation scale	The results show that sports intervention based on comprehensive theoretical foundation design is helpful to children’s physical and mental development.
Perić et al. (2021) [56]	Objective To explore the influence of adaptive football on down syndrome teenagers’ sports learning and some psychosocial characteristics.	25 males aged 15 to 17 years old	random	Intervention	Conduct special football training twice a week.	the Full Scale Intelligence Quotient (FSIQ)	Adaptive football training can reduce the aggressiveness of down syndrome teenagers and increase social behavior.
Jalilinasab et al. (2021) [57]	To examine the effects of Brain Gym exercise on development of fundamental motor and social skills.	84 children (average age: 9.55 years)	random	Intervention	16 brain training sessions	Matson Evaluation of Social Skills	Social skills are developed after Brain Gym exercise for children.

**Table 2 ijerph-19-06484-t002:** PEDro score.

Reference	Eligibility Criteria	Random Allocation	Concealed Allocation	Groups Similar at Baseline	Participants Blinded	Provider Blinded	Evaluator Blinded	Follow Up	Intention to-Treat Analysis	Between Group Comparison	PEDro Score
Samalot-Rivera et al. (2013) [32]	1	1	0	1	0	0	0	0	0	0	3
Pintérová et al. (2015) [33]	0	0	0	1	1	0	0	0	0	0	2
Metwaly et al. (2015) [34]	1	1	0	1	1	0	0	0	1	1	6
Sukys et al. (2016) [35]	1	0	0	1	1	0	1	1	1	1	7
Río et al. (2016) [36]	0	0	0	0	0	0	1	0	1	0	2
Folleto et al. (2016) [37]	1	0	0	0	0	0	1	1	1	0	4
Pan et al. (2016) [38]	1	1	1	1	1	0	0	0	1	1	7
Gorucu et al. (2016) [39]	0	1	1	1	1	0	0	0	0	1	5
Ferguson et al. (2016) [40]	1	0	0	0	0	0	0	1	1	0	3
Weiss et al. (2016) [41]	0	0	0	1	0	0	0	1	0	1	3
Phung et al. (2016) [42]	1	1	1	1	1	0	0	1	1	1	8
Lang et al. (2016) [43]	1	1	1	1	1	1	0	1	1	1	9
Bakır et al. (2017) [44]	0	0	0	1	1	0	0	0	0	1	3
Ruiz-Ariza et al. (2017) [45]	1	1	1	1	1	1	0	0	1	1	8
Lee et al. (2017) [30]	0	0	0	0	0	0	0	0	1	1	2
Malinauskas et al. (2017) [46]	0	0	0	0	1	0	0	0	0	1	2
Park et al. (2017) [47]	0	0	0	1	1	0	0	0	1	1	4
Messler et al. (2018) [48]	1	1	1	1	1	0	0	0	1	1	7
Gulati et al. (2019) [49]	1	0	0	1	1	0	0	0	1	1	5
Ryuh et al. (2019) [50]	1	1	1	1	1	0	0	0	1	1	7
Cai et al. (2020) [51]	1	1	1	1	1	0	0	0	1	1	7
Cai et al. (2020) [52]	1	1	1	1	1	0	0	1	1	1	8
Zwinkels et al. (2020) [53]	1	0	0	1	1	0	0	0	1	1	5
Ringenbach et al. (2020) [54]	1	0	0	1	1	0	0	0	1	1	5
Condello et al. (2021) [55]	1	1	1	1	1	0	0	1	1	1	8
Perić et al. (2021) [56]	1	0	0	1	1	0	0	0	1	1	5
Jalilinasab et al. (2021) [57]	1	1	1	1	1	0	0	1	1	1	8

## Data Availability

Not applicable.

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
