# Peer review of "Influence of Sports Activities on Prosocial Behavior of Children and Adolescents: A Systematic Literature Review"

_ijerph, 2022, doi:10.3390/ijerph19116484_

Round 1

Reviewer 1 Report

The manuscript describes a review of studies on interventions based on sport activities in the context of prosocial behavior. The goal of the study is important for social practice. The authors describe their searching process, inclusion and exclusion criteria, and summarized the results of the studies. Generally, I find the manuscript well-prepared. I have some minor suggestions.

#1. Please, try to indicate clearly that the goal was to analyse the interventions based od sport in your analysis. Please, justify also why the Authors did not analyse e.g. sport activity of the youth as a dispositional predictor.

#2. In my opinion, prosocial behavior (and its particular categories) should be described in the introduction. Prosocial behavior is a wide-range category, thus sport activity could be beneficial for some subcategories, while indifferent for others.

#3. In my opinion, the Table could also contain some effect sizes detected in the particular study. Moreover, the testing tools could also be slightly described i.e. in an appendix.

#4. I am wondering whether the analyses described in the 3.1. section are necessary. Given the low number of studies, region or date of publication could not be tested as moderating variable.

#5. I think that in the discussion the machanisms of sport-based interventions for prosocial behavior could be described in more details. It is important from the practical point of view.

Author Response

Response to Reviewer 1 Comments

Point 1: Please, try to indicate clearly that the goal was to analyse the interventions based on sport in your analysis. Please, justify also why the Authors did not analyse sport activity of the youth as a dispositional predictor.

The authors’ Answer: Many thanks for your kind comments. Revised. It has been stated in the introduction that "the research is based on the intervention measures of sports activities". In addition, there is no analysis of physical activity as a predictor of personality, because the evidence level of intervention research is higher than that of "analysis of physical activity as a disposition predictor (observational research)". Please read the details on page 1, line 14 and page 2, line 32 of the article.

Point 2: In my opinion, pro-social behavior (and its particular categories) should be described in the introduction. Pro-social behavior is a wide-range category, thus sport activity could be beneficial for some subcategories, while indifferent for others.

The authors’ Answer: Many thanks for your kind comments. Revised. The specific categories of pro-social behaviors have been described in the introduction. According to your suggestion, I have added that sports activities do not affect all indicators of pro-social behavior, and may be beneficial to some subcategories. Please read the details on page 1, line 29 of the article.

Point 3: In my opinion, the Table could also contain some effect sizes detected in the particular study. Moreover, the testing tools could also be slightly described in an appendix.

The authors’ Answer: Thank you for your proposal on the amount of effect. However, the expression of the effect quantity was not found in 24 original documents, and it did not explain the intervention effect in the original documents. It is more pertinent to evaluate the evidence level of the literature in Table 2 of Quality Evaluation. Because there is no specific index to evaluate the effect intensity.

I have described the testing tools in the appendix. Please read the details on page 17, line 13 of the article.

Point 4: I am wondering whether the analyses described in the 3.1. section are necessary. Given the low number of studies, region or date of publication could not be tested as moderating variable.

The authors’ Answer: Many thanks for your kind comments. Revised. Information, such as the year and region of publication of the literature, belongs to the basic information described in the literature. Although the number of the literature is small, the trend shown is still significant. This is based on the statistics made under the background that there is not much research on sports activities as children's pro-adolescent and pro-social behaviors around the world. To a certain extent, it reflects the attention of different regions to this issue.

Point 5: I think that in the discussion the mechanisms of sport-based interventions for pro-social behavior could be described in more details. It is important from the practical point of view.

The authors’ Answer: Many thanks for your kind comments. In the discussion sections 4.1, 4.2 and 4.3, a detailed description of the intervention mechanism of pro-social behavior has been added. Please read the details on page 15, line 2, line 35 and page 16, line 10 of the article.

Reviewer 2 Report

I just wanted to draw attention to the expression: "... normal children" (in the discussion of work, for example). Personally, I understand and I have no objection, but this form of expression can arouse some indignation in some people. I suggest you find another way to express yourself, such as children with special educational needs.

Author Response

Response to Reviewer 2 Comments

Point 1: I just wanted to draw attention to the expression: "... normal children" (in the discussion of work, for example). Personally, I understand and I have no objection, but this form of expression can arouse some indignation in some people. I suggest you find another way to express yourself, such as children with special educational needs.

The authors’ Answer: Many thanks for your kind comments. Normal children have changed to healthy children. We have changed children with disabilities to children with special educational needs. According to your suggestion, I also changed other expressions. Please read the details on page 14, line 14 of the article.

Reviewer 3 Report

Thank you for the opportunity to revise the manuscript entitled "Influence of Sports Activities on Prosocial Behavior of Children and Adolescents: A Systematic Literature Review"

The topic is very interesting, and the manuscript fits well with the Special Issue.

However, several points should be addressed.

Introduction section:

The Introduction is too schematic. Please rewrite the introduction in order to make it more readable.

"However, studies in China, the United States, Italy and other countries show that some children and adolescents are extremely lacking in prosocial behaviors[6,11-14]. Many problems caused by this, such as children's antisocial behavior, juvenile addiction behavior and juvenile delinquency, etc. It caused bad influence and harm to themselves, others and society."

I think these sentences are very extreme. I suggest mitigating them.

Please provide a more detailed explanation of the Prosocial-behaviour.

Methods section

Methods should be Method

- The search strategy should be ok; however, you reported studies with ADHD and down syndrome participants. In the Introduction, you mention the disabilities only at the end. I think if you want to include them, you should articulate more the introduction.

- I think most citations are misreported, e.g., Giancarlo Condello et al.(2021); you should write only the surname (i.e., Condello).

- Why did you select only manuscripts published between 2016 and 2021? It is not wrong, but you should explain that.

- "Avoid the subjective opinions, two reviewers will evaluate the opinions, and the third reviewer will judge the differences (Table2)."

Ok, Who were the reviewers? Were they experts in these kinds of research?

- "Author's region and country of origin. As shown in Figure 2B-C, the 24 relevant re- search papers included in this study are respectively from the United States, China, Korea, Turkey, Lithuania, Spain, Italy, the Netherlands, Switzerland, Serbia, Germany, Brazil, Iran and India. Geographically, Asian countries published 10 articles, accounting for 42% of the total, European countries published 9 articles, accounting for 38% of the total, North America published 4 articles, accounting for 17%, and South America published 1 article, accounting for 4% (Figuer1.b-c)."

I do not understand the importance of this. 

- "Sports Activities Can Improve the Prosocial Behavior of Normal Children and Adolescents".

I suggest changing the word Normal to Healthy

- In the Results section, somewhere, I also suggest reporting how you differentiated the various research (e.g., age, disabilities).

- The reason can be found from physiology. High-intensity interval training can increase cerebral blood flow and cate- cholamine production, which can improve children's and adolescents' self-efficacy, re- duce negative emotions, promote sustained entertainment and increase the occurrence of prosocial behaviors[64]. Especially when high-intensity activities are associated with co- operative activities, high-intensity sports activities in pairs or groups may be a new strat- egy to improve prosocial behavior[65]. Finally, highly structured stress training has also been found to develop a series of skills related to psychological resilience[51]. In adversity training, in order to overcome the psychological barriers deliberately set up in the adver- sity training camp, participants try their best to challenge their physiological needs and increase their confidence and ability to defeat their opponents. The attitude toward active response to challenges in the face of difficult environment can be the focus of children and adolescents' cultivation of psychological willfulness and social adaptability. In addition, stress training can also improve the way of thinking, enhance self-understanding and mo- tivation orientation, finally leading to changes in the personality, attitude and skills of participants in stress activities. In the future, through the innovation and integration of different types and natures of game activities, the relationship between sports activities and children's and adolescents' prosocial behaviors will be further elaborated, so that sports activities are expected to become an attractive scheme to improve children's and adolescents' prosocial behaviors.

Ok, it could be a possible explanation. Nothing about the cognitive and pedagogical perspective? 

Author Response

Response to Reviewer 3 Comments

Introduction section:

Point 1: The Introduction is too schematic. Please rewrite the introduction in order to make it more readable. I think these sentences are very extreme. I suggest mitigating them. Please provide a more detailed explanation of the Prosocial-behaviour.

The authors’ Answer: Many thanks for your kind comments. The introduction has been carefully revised. Although some sentences result from previous research, I have deleted and changed inappropriate sentences according to your suggestion. We have introduced pro-social behavior in detail. Please read the details on page 1, line 30 of the article.

Methods section

Point 2: Methods should be Method

The authors’ Answer: Many thanks for your kind comments. Revised. Please read the details on page 2, line 36 of the article.

Point 3: The search strategy should be ok; however, you reported studies with ADHD and down syndrome participants. In the Introduction, you mention the disabilities only at the end. I think if you want to include them, you should articulate more the introduction.

The authors’ Answer: In the introduction part, the description of children with special educational needs has been supplemented. Please read the details on page 1, line 39 of the article. Many thanks for your kind comments.We discuss the relationship between sports activities and pro-social behaviors of children and adolescents. Apart from healthy children and adolescents, "ADHD and Down syndrome children" belong to children with special educational needs, and they are also a group of children and adolescents involved in this research and should be included.

Point 4: I think most citations are misreported, e.g., Giancarlo Condello et al.(2021); you should write only the surname (i.e., Condello).

The authors’ Answer: Many thanks for your kind comments. All the citations were checked and changed. Please read the details on the table on page 4 of the article.

Point 5: Why did you select only manuscripts published between 2016 and 2021? It is not wrong, but you should explain that.

The authors’ Answer: Many thanks for your kind comments. Before 2016, there were very few articles about sports activities as evidence to influence children's pro-social behavior intervention, which were basically published after 2016. In order to ensure the timeliness of the literature, we included articles from 2016 to 2021. Please read the details on page 2, line 42 of the article.

Point 6: "Avoid the subjective opinions, two reviewers will evaluate the opinions, and the third reviewer will judge the differences (Table2)."Ok, Who were the reviewers? Were they experts in these kinds of research?

The authors’ Answer: Many thanks for your kind comments. The reviewers are Professor Shao and Professor Qi, who are experts in this field.

Point 7: "Author's region and country of origin. As shown in Figure 2B-C, the 24 relevant re- search papers included in this study are respectively from the United States, China, Korea, Turkey, Lithuania, Spain, Italy, the Netherlands, Switzerland, Serbia, Germany, Brazil, Iran and India. Geographically, Asian countries published 10 articles, accounting for 42% of the total, European countries published 9 articles, accounting for 38% of the total, North America published 4 articles, accounting for 17%, and South America published 1 article, accounting for 4% (Figuer1.b-c)." I do not understand the importance of this. 

The authors’ Answer: Many thanks for your kind comments. Information, such as the year and region of publication of the literature, belongs to the basic information described in the literature. Although the number of the literature is small, the trend shown is still significant. This is a statistical icon based on the background that there are few related studies around the world, and to some extent, it reflects the attention of different regions to this issue.

Point 8: "Sports Activities Can Improve the Prosocial Behavior of Normal Children and Adolescents".

I suggest changing the word Normal to Healthy

The authors’ Answer: Many thanks for your kind comments. Normal children have changed to healthy children. Please read the details on page 13, line 20 of the article.

Point 9: In the Results section, somewhere, I also suggest reporting how you differentiated the various research (e.g., age, disabilities).

The authors’ Answer: Many thanks for your kind comments. In the results part, we have added that the research results are distinguished according to the difference of the influence of different sports activities on pro-social behaviors and the difference of the influence of sports activities on physical function. Please read the details on page 13, line 16 of the article.

Point 10:The reason can be found from physiology. High-intensity interval training can increase cerebral blood flow and cate- cholamine production, which can improve children's and adolescents' self-efficacy, re- duce negative emotions, promote sustained entertainment and increase the occurrence of prosocial behaviors[64]. Especially when high-intensity activities are associated with co- operative activities, high-intensity sports activities in pairs or groups may be a new strat- egy to improve prosocial behavior[65]. Finally, highly structured stress training has also been found to develop a series of skills related to psychological resilience[51]. In adversity training, in order to overcome the psychological barriers deliberately set up in the adver- sity training camp, participants try their best to challenge their physiological needs and increase their confidence and ability to defeat their opponents. The attitude toward active response to challenges in the face of difficult environment can be the focus of children and adolescents' cultivation of psychological willfulness and social adaptability. In addition, stress training can also improve the way of thinking, enhance self-understanding and mo- tivation orientation, finally leading to changes in the personality, attitude and skills of participants in stress activities. In the future, through the innovation and integration of different types and natures of game activities, the relationship between sports activities and children's and adolescents' prosocial behaviors will be further elaborated, so that sports activities are expected to become an attractive scheme to improve children's and adolescents' prosocial behaviors.

Ok, it could be a possible explanation. Nothing about the cognitive and pedagogical perspective? 

The authors’ Answer: Many thanks for your kind comments. In the discussion section, we have supplemented the viewpoints of cognition and teaching. Please read the details on page 15, line 34 of the article.

Round 2

Reviewer 3 Report

Dear authors

Thank you for your detailed responses.

the Introduction section does not fully convince me. It needs extensive editing of English and the sentences should be linked in a more appropriate way.

Please, explain more in-depth the prosocial behaviors: sharing, helping and comforting.

The authors’ Answer: Many thanks for your kind comments. Before 2016, there were very few articles about sports activities as evidence to influence children's pro-social behavior intervention, which were basically published after 2016. In order to ensure the timeliness of the literature, we included articles from 2016 to 2021. Please read the details on page 2, line 42 of the article.

I don't see the problem to perform the systematic search with a starting year before 2016. If there are on few articles, I think you should report them.

The authors’ Answer: Many thanks for your kind comments. Information, such as the year and region of publication of the literature, belongs to the basic information described in the literature. Although the number of the literature is small, the trend shown is still significant. This is a statistical icon based on the background that there are few related studies around the world, and to some extent, it reflects the attention of different regions to this issue.

Ok, thus I expect that this should be discussed in the Discussion or Conclusion section. Perhaps you may suggest future research.

The other points seem ok.

Author Response

Point 1: The Introduction section does not fully convince me. It needs extensive editing of English and the sentences should be linked in a more appropriate way.

Please, explain more in-depth the pro-social behaviors: sharing, helping and comforting.

The authors’ Answer: Many thanks for your kind comments. I have made targeted supplements to the introduction. I have introduced prosocial behavior in detail. I have revised the English according to your suggestion. Please read the details on page 1, line 26 of the article.

Point 2: I don't see the problem of performing the systematic search with a starting year before 2016. If there are on few articles, I think you should report them.

The authors’ Answer: Many thanks for your kind comments. Revised. I have re-searched all the articles, and the time limit was removed in the process of searching. A total of 27 articles were got after searching. I added 3 articles of intervention research that fit the theme: 1 article in 2013, 2 articles in 2015. Please read the details on page 4, Table 1. Study Characteristics.

Point 3: Thus I expect that this should be discussed in the Discussion or Conclusion section. Perhaps you may suggest future research.

The authors’ Answer: Many thanks for your kind comments. Revised. I have added this part to the discussion part. I have explained it in the future outlook. Please read the details on page 14, line 36 of the article.